# The Power of Collective Design: Co-Creating Healing-Centered Mental Health Care for Refugee and Immigrant Families

**DOI:** 10.3390/ijerph22071035

**Published:** 2025-06-28

**Authors:** Reba Meigs, Adriana Bearse, Amina Sheik Mohamed, Sarah Vicente, Arwa Alkhawaja, Ariana Aini, Wali Abdul Hanifzai, Gulshan Yusufzai, Sara Mostafavi, Ruth Teseyem Tadesse, Reem Zubaidi, Mohammad Wahdatyar, Nghi Dang, Asmaa Deiranieh, Segen Zeray, Farhat Popal, Valerie Nash, Blanca Melendrez

**Affiliations:** 1Center for Community Health, University of California San Diego, La Jolla, CA 92093, USA; asheikmohamed@health.ucsd.edu (A.S.M.); shvicente@health.ucsd.edu (S.V.); rtadesse@health.ucsd.edu (R.T.T.); rnzubaidi@health.ucsd.edu (R.Z.); mwahdatyar@health.ucsd.edu (M.W.); npdang@health.ucsd.edu (N.D.); adeirani@health.ucsd.edu (A.D.); szeray@health.ucsd.edu (S.Z.); f1popal@health.ucsd.edu (F.P.); bmelendrez@health.ucsd.edu (B.M.); 2Rihla Community Services, San Diego, CA 92126, USA; maenarwa@gmail.com; 3International Affairs, San Diego State University, San Diego, CA 92182, USA; 4Maristan, Fremont, CA 94539, USA; aini@maristan.org (A.A.); smostafavi@maristan.org (S.M.); 5Qazizada Multicultural Therapy Clinic, Orange, CA 92867, USA; wali31950@gmail.com; 6MAS Social Services Foundation, Sacramento, CA 95821, USA; gulshan.yusufzai@mas-ssf.org; 7Nash and Associates, San Diego, CA 92102, USA; valerienash5818@gmail.com

**Keywords:** trauma-informed care, healing-centered, social determinants of health, community-led transformation, collective impact, community resilience, cultural sensitivity, refugee health, immigrant health

## Abstract

Refugee and immigrant communities face a host of dynamic health challenges. This essay discusses the importance of prioritizing the impacts of resettlement on mental health and provides examples of how creating a collective network of culturally responsive, trauma-informed, and healing-centered providers—centered on community-based best practices and knowledge—is integral to fostering community resilience. Additionally, it will examine how resettlement challenges intersect with systemic barriers to culturally responsive care and related Social Determinants of Health (SDOH), including economic stability, health care access and quality, and social and community contexts. Drawing upon experiences from a statewide network spanning southern and northern California—and based on interim mixed-methods program evaluation data and practitioner reflections highlighting one community partner’s experience—we provide key learnings that demonstrate how coalition building, cultural humility, and provider training can improve client well-being, reduce mental health disparities, and address the relevant SDOH. Key learnings illustrate the importance of the following: (1) coalition building to co-create trusted provider referral networks and support peer-to-peer learning to enhance client care; (2) provider trainings and capacity building on healing-centered and culturally responsive practices to address SDOH; (3) centering cultural humility; and (4) building a peer-based workforce who speak similar languages and share lived experiences to provide deeper cultural connections and build trust. Our experiences demonstrate that the co-creation of strong mental health provider networks is critical to strengthening the fabric of community resilience.

## 1. Introduction

Refugee and immigrant communities face trauma not only from forced displacement and resettlement but also from systemic barriers shaped by policies that limit access to essential resources. Restrictive immigration policies, legal uncertainty, and limited healthcare access compound the stress and mental health challenges many communities experience. Without adequate support, these structural barriers further deepen health disparities [1].

Despite these challenges, communities have built powerful networks of care, rooted in cultural resilience and mutual aid. As public health continues to face tremendous challenges, community-based organizations (CBOs) are leading collective action efforts to mitigate the impact of systemic barriers, foster trust, and address Social Determinants of Health (SDOH) [2] by creating networks of culturally responsive providers who center community-based best practices, knowledge, and power [3]. In this paper, we prioritized discussion of specific SDOH, such as economic stability, healthcare access and quality, and social and community contexts based on the following three key factors: (1) direct input from community partners and CBOs during co-design processes; (2) findings from regional needs assessments and evaluation data; and (3) alignment with the capacities of partners to deliver culturally and linguistically aligned interventions.

Building on the Community-Led Transformation [4] approach as a foundation to our program, this paper demonstrates how coalition-building, healing-centered practices, cultural humility, and hiring culturally sensitive staff are integral components of successful community-led programs that support the mental health and well-being of refugee and immigrant communities. We describe the community-driven, collective impact approach to developing and implementing the Afghan Refugee School Impact (ARSI) and Afghan Youth Mentoring (AYM) programs in collaboration with over 40 CBOs throughout California, including four geographic regions—San Diego, Los Angeles/Orange/Inland, Bay Area San Francisco, and Sacramento/Central Valley. We also provide a case study example of one participating mental health provider organization’s experience and key learning points derived from serving ARSI/AYM families in the Bay Area. In addition, we share specific recommendations that funders, providers, and CBOs can use to successfully co-design healing-centered strategies that cultivate environments of belonging and trust, addressing relevant SDOH through culturally rooted knowledge and shared leadership [5].

Afghan Refugee School Impact (ARSI) and Afghan Youth Mentoring (AYM) Programs. In 2022, the statewide ARSI and AYM programs were established with $13 million in funding from the California Department of Social Services, using federal resources from the Office of Refugee Resettlement (ORR) under the Afghanistan Supplemental Appropriation, 2022, and Additional Afghanistan Supplemental Appropriations Act, 2022 (ASA), spanning four regions across southern and northern California. Working in collaboration with the CDSS Office of Immigrant Youth, ARSI/AYM brings together a statewide network of over 40 CBOs that provide a range of direct services and support to youth and families newly arrived from Afghanistan [6]. The overarching goal of ARSI/AYM is to engage and connect youth and families with local services that are culturally responsive and integrate trauma-informed and healing-centered practices that are based on regional and local needs.

Since its inception, the ARSI/AYM program has served over 3000 youth and family participants, including culturally responsive mental health services, such as peer-based support groups, nonclinical counseling, social–emotional workshops, health and hygiene education, and anti-bullying programs for approximately 800 refugee youth and families.

Community-Based Mental Health Provider: Maristan. One of the CBOs serving as a key local partner in this work is Maristan, a mental health provider organization in the San Francisco Bay Area whose team provides culturally sensitive healing-centered direct services, including mental health support, to a diverse group of families. Building on the traditional Islamic purpose of the Maristans and functioning as the community partner of the Muslim Mental Health & Islamic Psychology Lab at Stanford University [7], Maristan’s staff use research and education to inform holistic healing for their clients and provide culturally and spiritually congruent, professional, accessible, affordable mental health care for all.

Maristans were early institutions of healing in the Islamic world, established primarily to treat mental health conditions. Beginning in the 8th century, they played a pivotal role in the development of psychiatric care within Islamic civilization. These institutions were innovative in integrating mental health treatment into the broader healthcare system. They often operated as part of larger hospital complexes or as standalone facilities specifically dedicated to the care of the mentally ill. Maristans were grounded in core values such as compassion, cultural sensitivity, holistic well-being, and accessibility, offering specialized care that was advanced for its time and laid the foundation for modern mental health practices [7].

## 2. Methods

The ARSI and AYM programs employ a mixed-methods evaluation approach to measuring outcomes at the community and systems levels and identifying key learnings across the state. Preliminary evaluation data collected to date [8] include quantitative and qualitative data from participant surveys (sample of *n* = 200 out of over 3000 program participants) and stories; partner agency surveys (*n* = 40 out of 47 partner CBOs), focus groups, and semi-structured interviews (*n* = 7), along with ongoing practitioner reflections captured through Community of Practice trainings and communications with the program’s over 40 CBO partners and practitioners—including Maristan—to inform identification of lessons learned and continuous quality improvement across program implementation. Analysis methods include descriptive analyses of quantitative participant data and closed-ended survey responses from participant and partner survey questions, and thematic analysis and integration of qualitative data collected via focus groups, interviews, and participant stories to identify key themes. This evaluation approach reflects our commitment to integrating applied evaluation findings triangulated from multiple sources, while honoring the lived experiences of both participants receiving services and practitioners conducting service delivery, in line with the participatory framework utilized throughout ARSI/AYM.

## 3. Key Strategies and Learnings

The following sections describe four key strategies employed by ARSI/AYM and Maristan to address SDOH for refugee youth and families, highlighting learnings from interim evaluation data collected to date and practitioner reflections on key factors that have facilitated success. These strategies and learnings illustrate healing-centered work in action from a statewide network of ethnic-based community organizations.

### 3.1. Coalition Building and Establishing Trust

The collective impact framework highlights the transformative power of collaboration in addressing complex social challenges, amplifying impact by fostering inter-organizational partnerships and pooling resources [9,10]. Central to this framework is coalition building, which enables diverse stakeholders to co-create culturally responsive and healing-centered solutions that enhance resilience and address the SDOH and health challenges faced by refugee and immigrant communities [11,12,13]. The development of trusted partnerships and collaboration may directly impact the SDOH, including health care access and quality [14].

Coalition building is instrumental in developing trusted referral networks and supporting peer-to-peer learning [10]. These networks facilitate collaboration among culturally responsive, healing-centered providers equipped to meet the unique needs of vulnerable communities [15]. Fostering trust is an essential aspect of healing centered practices and has a direct impact on the health of the community and its economic viability [12]. This trust means agreements are kept, effort is made to show up, and community is included in the decision-making process. Establishing trust takes vulnerability and the discernment to approach situations with humility and sensitivity.

A strong, experienced backbone organization is key to coalition building and establishing trust to achieve a collective impact approach. With the University of California San Diego (UC San Diego) Center for Community Health (CCH) Refugee and Immigrant Health (RIH) unit serving as the backbone organization, ARSI/AYM has established four regional coalitions to facilitate capacity building and partnership development among CBOs across the state. The foundational coalition in ARSI/AYM is the long-standing San Diego Refugee Communities Coalition (SDRCC)—comprising approximately a dozen ethnic-led community organizations across the region—based on which the other three ARSI/AYM regional coalitions were modeled at the outset of the program [16]. The ARSI/AYM coalition model includes the implementation of regional and statewide Communities of Practice (CoP) for participating organizations, providers, and staff, focused on building capacity and providing opportunities for peer learning across topics ranging from cultural humility and healing-centered practices to youth development to grant writing and data collection. As a function of the backbone organization, CCH utilizes a statewide communication technology platform to enhance the sharing of information among all four coalitions. The ARSI/AYM partners utilize Basecamp, a communication tool that allows the users to post messages, share resources, seek advice from peers on best practices, and promote events. Additionally, each statewide coalition has established a WhatsApp group to foster real-time, on-the-ground, responsive communication to address the specific needs of clients in their region.

Preliminary ARSI/AYM evaluation data collected through surveys, focus groups, and interviews with participating organizations shows evidence of the program’s early success in achieving organizational- and systems-level outcomes, with findings indicating increased organizational capacity; increased collaboration across CBOs in each region; and CBO plans to sustain partnerships developed through ARSI/AYM [8]. By fostering the exchange of best practices and addressing service gaps, statewide coalitions like ARSI/AYM can improve the quality of care and strengthen the resilience of community-based mental health systems [12]. Partnerships, often built or nurtured through coalitions, further enhance these efforts to effectively address SDOH—including healthcare access as well as social and community context factors such as health literacy—by leveraging resources and fostering culturally sensitive care approaches [11].

### 3.2. Building Capacity in Healing-Centered Care

While trauma-informed care (TIC) is widely recognized and used in refugee mental health, it often centers deficits rather than strengths, focusing on past trauma instead of community resilience and cultural healing practices [3,14]. TIC asks, “What happened to you?” but does not always ask, “What strengths and cultural assets support your healing?” This approach risks medicalizing refugees’ experiences rather than addressing key SDOH, such as economic stability and access to healthcare.

In contrast, healing-centered engagement (HCE) shifts the focus to holistic well-being, identity, and empowerment. By embedding community-led, culturally affirming approaches, HCE ensures that refugee mental health interventions are not just clinically effective but also culturally rooted and sustainable. The approach highlighted here builds upon trauma-informed principles by advocating for a more strengths-based and holistic healing-centered approach, which shifts the focus from treating trauma to fostering well-being, connection, and resilience [4].

To foster best practices that emphasize co-created solutions prioritizing healing-centered care, it is important to provide capacity building and workforce development opportunities to direct service providers. Capacity building and workforce development are necessary components to addressing economic stability, an identified domain of the SDOH [2]. Programs like ARSI/AYM demonstrate healing-centered practices through strategies such as training providers in healing-centered care and hiring culturally responsive staff to foster trust, resilience, and self-determined healing, which in turn helps reduce barriers to care, increase access to health services, and contribute to economic stability for providers and patients. Preliminary evaluation results from ARSI/AYM participant surveys and stories indicate that these services have successfully led to improved access to mental health care through cross-agency collaboration and peer-to-peer referral networks, along with participant outcomes, including self-reported improvements in parent and youth coping skills to deal with stress and trauma; family communication skills and relationships; and self-reported social and emotional health and well-being [8].

To achieve these outcomes, the ARSI/AYM program fosters capacity building using CoP as one method for implementing a statewide professional development and collaboration model. The CoP focuses on providing training that is healing and culturally centered, including training specifically focused on the use of healing-centered practices in care. The ARSI/AYM evaluation data highlight the value of the CoP training provided through the program, with approximately 9 out of 10 participants indicating that they learned something new; that they gained information/skills to support their communities; and that the CoP format provided opportunities for them to engage in peer learning and collaboration with others also serving Afghan youth and families [8]. CoP training specifically focused on trauma-informed and healing-centered practices were amongst those most highly rated in terms of usefulness. ARSI/AYM evaluation data from surveys, focus groups, and interviews also shows the program’s role in strengthening CBO infrastructure and expanding services and capacity, highlighting key program aspects such as training, technical assistance, and regional partnerships as critical to fostering trust and improving service coordination to meet community needs [8].

### 3.3. Centering Cultural Humility

Utilizing culturally responsive approaches when working with refugee and immigrant populations to address SDOH—including social and community contexts and health care access and quality—is key to establishing a long-term client-centered relationship. Cultural humility, which is the “process of self-reflection and discovery in order to build honest and trustworthy relationships,” is a crucial component of any public health initiative, possibly even more so when working with refugee and immigrant families [17]. As one among a set of values that we believe should guide the collaborative work of addressing these SDOH, cultural humility ensures that we strive to deeply understand and honor the rich and diverse backgrounds of the individuals, families, and communities we serve, recognizing the pivotal role of our community partners as cultural stewards. Cultural humility demands that we prioritize patience, empathy, and care while navigating diverse perspectives, and that we are careful to identify our own biases and assumptions so as not to impose them on the communities we serve. It requires meeting the needs of the clients based on their terms, and it can have a direct impact on health literacy and health care access and quality.

The ARSI/AYM program works to build a system that is culturally responsive to the customs, practices, and holidays of our partners and participants. For example, when determining programmatic timelines, we discussed the importance of not starting or ending programs during the holy month of Ramadan, as well as the need to be flexible with meeting scheduling and data deadlines to ensure that staff at all levels of the program—including university backbone staff, regional leads (RLs), and sub-awardee organizations (SAs)—are able to participate in their faith traditions while meeting the needs of the program. We also know that the RLs and SAs may have to prioritize community needs in addition to those from the ARSI/AYM program. To support participants, we encourage organizations to address the most urgent needs with flexibility, provided they remain within contractual requirements.

The ARSI/AYM CoP model has also directly addressed cultural humility. At the outset of our second round of programming, we hosted a training focused broadly on cultural humility and specifically on Afghan culture. This session was led by an Afghan member of the UC San Diego backbone team, who taught participants about the customs, practices, and holidays observed by many ARSI/AYM participants, following which 90% of participants indicated increased confidence in their ability to promote cultural humility within the ARSI/AYM program, and 95% reported feeling better prepared to effectively engage and outreach to Afghan communities [8].

### 3.4. Building a Peer-Based Workforce

Ensuring that the work environment and communication align culturally with the value system of the client is foundational in delivering effective mental and behavioral health services. Creating spaces where clients feel comfortable and are able to express their challenges freely is the goal of healing-centered practice. One way to support this type of authentic interaction is by prioritizing the hiring of staff who speak the language(s) of clients and their families. When program participants are able to communicate in their native language, they are able to connect and engage in authentic ways [18]. Hiring staff with lived experiences that reflect their communities also enhances cultural connections and effectively addresses systemic barriers [19]. In addition, staff who can communicate with clients in their primary language and bring lived experiences to their work enhances participant trust, engagement, and retention in services [5]. Preliminary evaluation data collected through ARSI/AYM participant and partner surveys, focus groups, and interviews indicates high participant satisfaction with services, highlighting the value of peer- and group-based approaches as particularly effective in fostering trust, social connections and community for participating youth and families [8]. Overall, hiring peer-based staff addresses multiple SDOH, including economic stability through job creation as well as health care access and quality through language-aligned care [5].

Within the ARSI/AYM programs, providers were assessed to determine the extent to which they represented the community to be served. UC San Diego made focused efforts to identify Afghan-led organizations that met the criteria as an Ethnic Community-Based Organization [20] and invited them to participate in the ARSI/AYM program, including those that were newly created and did not have a history of operating government-funded services. Other partners were asked to describe their experience and capacity to serve Afghan families, as well as to indicate how many Afghan staff they employed. During the co-creation of program designs, the need to ensure that direct services were delivered by individuals who were fluent in Dari and/or Pashto, as well as familiar with the cultural norms and practices of the Afghan community, was emphasized.

UC San Diego CCH’s RIH team also incorporated the following practices into the ARSI/AYM design and delivery of CoP training, based upon learnings gained through experiences with SDRCC:Engagement of frontline staff in design meetings to incorporate their ideas and build their understanding of program goals and requirements;Inclusion of facilitated peer-based discussions and other interactive learning approaches within CoP trainings;Establishment of multiple avenues for accessing technical assistance and support with operational aspects of the program (fiscal management, data collection, human resources); including office hours, distribution of guidelines and protocols, and regular meetings and communications leveraging partner-preferred technology applications, such as WhatsApp.

In addition to working with partner organizations that represent the communities they serve, CCH has also remained committed to hiring staff that share backgrounds and lived experiences with the refugee and immigrant communities served through projects like ARSI/AYM. Over three-quarters of CCH staff self-identify as non-white; close to half were born outside the U.S.; and approximately two-thirds report having a parent or guardian who came to the U.S. as a refugee, immigrant, or asylee. Having a backbone staff with these shared backgrounds and lived experiences has helped enhance the integration of culturally responsive approaches throughout implementation of ARSI/AYM.

## 4. Case Study Example: Maristan

Mental health provider Maristan was one of the ethnic-based CBOs to participate in the statewide ARSI/AYM program, serving approximately 50 families—including over 100 youth and parent participants—through ARSI/AYM [8]. The following case study highlights Maristan’s experiences in applying the key strategies and learnings highlighted above, providing examples from the field that demonstrate how these learnings can be effectively employed to address SDOH—including health care access and quality, economic stability, and social and community contexts—for refugee and immigrant communities. The information in this case study draws upon testimonials and practitioner reflections from the staff and leadership at Maristan, who bring extensive lived experiences and employ a healing- and client-centered, trauma-informed approach to working with immigrants and refugees.

Building Capacity in Healing-Centered Care through Training: Maristan therapists undergo inclusive training and continuous education, which allows them to keep up with cultural trends and apply the most up-to-date practice. This can be seen in our one-on-one sessions with Afghan families during co-events with our coalition partners. During these events, some family members request services that align with their cultural beliefs, and our clinicians are able to create a safe space where their cultural identity is preserved and respected. They also lead support group sessions in Dari and bring pastries from Afghanistan. Most of the support groups have spare time during which the mothers and clinicians would engage in activities and craft sessions originating from Afghanistan. These activities consist of making Afghan bracelets or embroideries, which are then given as gifts to the other members of the support groups.

Centering Cultural Humility: Maristan offers virtual coaching sessions, conducted by highly trained Dari-speaking therapists (licensed clinical social workers, registered family therapists), which focus on empowering youth and families with essential life skills, academic support, and cultural adaptation strategies. These one-on-one coaching sessions offer personalized mental health support, trauma-informed care, and crisis intervention as needed. Sessions are scheduled to accommodate family needs, ensuring consistent engagement and progress tracking. Maristan support group sessions (also virtual) are held monthly and provide a safe space for Afghan youth and adults to connect, share experiences, and access peer support. These groups facilitate resource navigation and community networking.

Building a Peer-Based Workforce through Hiring Culturally Sensitive Staff: Maristan’s program not only seeks to provide mental health support but also to establish services that are culturally aligned and sensitive to the needs of the Afghan refugee community. As a team, we recognize that hiring staff that lack an understanding of the struggles of Afghan refugees or their traditions would undermine the effectiveness of the program. Therefore, we hire team members who understand Afghan culture and values, are fluent in Dari or Pashto, and are trained to understand the process of displacement and resettlement and how that can psychologically impact Afghan refugees. By hiring culturally sensitive staff, our team members are able to build trust with the Afghan families, resulting in individuals being more open to sharing their concerns, as they feel understood at both the linguistic and cultural levels. For example, our staff have conducted outreach events with Afghan families who were hesitant to participate when services were discussed only in English. However, once communication was conducted in Dari, the families were more enthusiastic to ask questions and participate. Our staff even shared their stories and personal experiences from Afghanistan, creating a more lasting therapeutic alliance and a higher rate of interest among the families. In addition, to stay connected, responsive, and aware of community needs, our staff participate in the ARSI/AYM WhatsApp Group. Through this platform, our staff have access to a network of peer-based providers to ensure that culturally appropriate resources are provided to the clients in a timely manner.

## 5. Call to Action

A collaborative spirit allows stakeholders to pool resources and align their efforts toward shared objectives, such as enhancing mental health services for populations with unique challenges [11,12]. Funders often achieve a better return on investment when initiatives prioritize collaboration [21]. A shared vision and accountability fostered through coalition building enhance the impact and sustainability of these efforts [22]. An asset-based approach to collaboration ensures that partnerships are grounded in cultural sensitivity and humility, recognizing and valuing the strengths of each contributor [10,23].

Organizational support is critical for establishing infrastructure to implement healing-centered and culturally sensitive care practices. These principles are fundamental when addressing the mental health needs of refugee and immigrant populations, who often face unique challenges related to resettlement, trauma, and systemic inequities [21].

To truly support the well-being of refugee and immigrant communities, we must shift from trauma-informed models to healing-centered systems that prioritize resilience, cultural strengths, and community leadership. This requires sustained investment, policy alignment, and institutional commitment. As such, we call on funders, policymakers, healthcare institutions, and CBOs to take the following actions:Invest in Healing-Centered Initiatives—Fund community-led programs that integrate culturally rooted healing practices, peer-led mental health support, and interventions designed to address SDOH;Embed Healing-Centered Care in Policy—Ensure that public health and behavioral health policies recognize culturally affirming, strengths-based models as essential to refugee and immigrant health;Strengthen Community-Led Workforce Development—Expand hiring and training for culturally responsive providers, peer navigators, and community-based mental health specialists;Build Sustainable Coalitions—Strengthen cross-sector partnerships, trusted referral networks, and regional Communities of Practice to drive systemic change;Shift Power to Community Leadership—Support participatory grantmaking and co-designed initiatives where refugee and immigrant communities lead program design and decision-making.

## 6. The Role of Technology in Delivering Culturally Responsive Refugee and Immigrant Health Services

Additionally, as new digital platforms and technological advances increasingly influence healthcare delivery and communications, integrating these tools into mental health services for refugee and immigrant communities can serve to further strengthen an initiative’s ability to be culturally responsive. Technology tools such as community-friendly communication platforms (e.g., WhatsApp), language translation software, digital software to support mental health screenings, and predictive analytics hold promise for improving access to care and can be thoughtfully implemented with ethical safeguards to ensure that these technologies do not inadvertently reinforce cultural biases, misinterpret distress signals, or fail to address the complex SDOH that shape refugee well-being [17]. Technology tools can be leveraged as tools for equity rather than exclusion through the following:Engaging community leaders and practitioners in selection and design of technology tools to build trust and ensure real-world applicability;Ensuring any technology tools used are culturally responsive and reflect the lived experiences of refugee communities;Safeguarding data privacy and security, as well as recognizing that many refugees have concerns about digital surveillance and potential misuse of their personal information.

When used appropriately, technology can be ethically integrated and leveraged within community-led mental health initiatives to help public health organizations improve cultural responsiveness and maximize human-centered care.

## 7. Limitations

It is important to consider that some of the outcomes resulting from the specific programs described here may be in part due to the time and trust built with partners in co-designing and implementing the ARSI/AYM project, which may limit the generalizability and scalability of this type of program across other organizations who may not have the extensive time, resources, and networks needed to employ the co-design and trust-building approach described herein.

Additionally, these programs are still in progress and thus final outcome data are not yet available. It is possible that some of the implementation methodologies may not be as effective as we believe them to be at this point of the program, based on the preliminary data available at this time. Further, as the programs described are specifically for Afghan individuals and communities, we cannot say that the findings are necessarily applicable to all immigrant or refugee communities.

Future evaluation and research could focus on more rigorous data collection and analysis to support the conclusions drawn herein and further expand the literature on the effectiveness of community-based, co-designed programs for immigrants and refugees.

## 8. Conclusions

While systemic barriers continue to impact the mental health and well-being of refugee and immigrant families, the power of community-driven solutions cannot be overstated. The collective impact approach—centered on coalition-building, cultural humility, and peer-to-peer learning—demonstrates how communities can lead the way in filling gaps left by restrictive policies and fragmented healthcare systems. Investing in these networks of care is essential to ensuring long-term resilience, equity, and healing for displaced populations.

As described throughout this paper, coalition building, cultural humility, and collaborative partnerships are essential for enhancing mental health services and addressing SDOH—such as health care access and quality, economic stability, and social and community contexts—in refugee and immigrant communities. Through fostering coalitions built on these values, stakeholders co-create solutions that strengthen resilience and improve mental health outcomes [10]. These efforts demonstrate the importance of statewide infrastructure in creating trusted networks and fostering peer-to-peer learning. By adopting collaborative and asset-based approaches, stakeholders develop culturally sensitive and healing-centered solutions that enhance community resilience and ensure sustainability. By aligning funding, policy, and workforce strategies with healing-centered approaches and incorporating innovative strategies, such as ethical technology integration, we can create lasting and equitable change, ensuring that communities not only recover but thrive.

## Data Availability

The data presented in this article are not readily available because they were collected for the ARSI/AYM program for program evaluation purposes only [see reference [8] for interim program evaluation report] and were not consented for research use or public sharing. Requests regarding this data should be directed to the corresponding authors.

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
