# Peer review of "The Power of Collective Design: Co-Creating Healing-Centered Mental Health Care for Refugee and Immigrant Families"

_ijerph, 2025, doi:10.3390/ijerph22071035_

Round 1

Reviewer 1 Report

Comments and Suggestions for Authors

This report adds to the literature on culturally appropriate services. Although the program is unique to the Afghan community, it can be adapted for use in other immigrant communities. Please see my comments throughout the document. 

Reviewer 2 Report

Comments and Suggestions for Authors

This is a very well written article that articulates its purposely clearly. The only challenge I found while reading it was wondering about the geographic locations of each of the programs. It seems that each program may be at a different locations (UCSD) but it was not made clear. It highlights the importance of community engagement and be sensitive to community needs.

 There are very minor concerns in regards to this article:

*       A good part of this article addresses Maristans, but it is not mentioned in the abstract or introduction section. This should be added to accurately show contents of this article.
*       ABSTRACT: Is there any geographic information that can be added regarding where the programs are located? It could assist readers organize and understand the article better.
*       Line 91-99 gives historical context of Maristans. The purpose of this paragraph needs to be tied in a more clearly way to the overall article.
*       Line 253: the language of case study is used. Traditional case studies usually exemplify a single case in detail. Perhaps this is not a "case study section" but a "process of Maristan". Leaving the language of case study is mis leading.

Author Response

See attached pdf. 

Reviewer 3 Report

Comments and Suggestions for Authors

This essay investigates how healing-centered, culturally responsive mental health care that is co-designed with refugee and immigrant communities can help address socioeconomic determinants of health and minimize mental health inequities. Drawing on a case study of Maristan and statewide coalition-building initiatives in California, it proposes meaningful strategies to promote community resilience. Here are some key points for authors to consider in order to enhance their manuscript.

  1. TITLE: The current title is self explanatory and descriptive. However, it is length and philosophical richness may obscure its clarity.  
  2. ABSTRACT: The abstract outlines “key learnings” and “examples” but does not specify how they were produced (e.g., evaluation data, interviews, case studies, practitioner reflections). The authors should describe how the key learnings are drawn for the benefit of the reader and for the credibility of these findings.
  3. INTRODUCTION: Line 42-45. The authors mention addressing Social Determinants of Health (SDOH). However, which specific SDOH are targeted should be specified for the benefit of the reader. The abstract mentions broad terms like “economic instability” and “limited healthcare access.” The Introduction could benefit from a better articulation of the specific SDOH being targeted. They should ideally be aligned with Healthy People 2030 domains like economic stability, housing, education, healthcare access, neighborhood environment, and social context, but authors can explain why the specific SDOH are selected.
  4. In addition, the SDOH indicated in the Introduction seems to contradict the four essential approaches outlined in the essay. Coalition building and cultural humility enable SDOH but are not SDOH. To improve clarity and conceptual alignment, the essay should explicitly relate these tactics to the SDOH they want to influence.
  5. It will be useful to include a brief subsection covering methodology to derive "key learnings" and how they were validated. The reader may benefit from details such as whether surveys, focus groups, or anecdotal observations were used to draw conclusions.
  6. The authors should reconsider combining the Introduction and Background sections into one; otherwise, the decision to place certain materials separately into two sections seems arbitrary.
  7. In addition, most of the background section is about advocacy without defining a clear description of the knowledge gap.
  8. The description of each of the four "Key Strategies and Learnings" seems conceptual without sufficient data-driven evidence. The authors should consider addressing this.
  9. Although the case study on Maristan is useful, it could be more closely related to the larger essay by systematically discussing how it illustrates the main techniques instead of simply showing anecdotal evidence.
  10. The authors should list any limitations in a paragraph or a section. They should discuss whether these key strategies and lessons learned are applicable in other settings. To what extent are they generalizable?
  11. The authors should more explicitly distinguish between trauma-informed care and healing-centered engagement in a comparative manner. This is important given that this difference is a main conceptual contribution of the article.
  12. Conclusions: Rather than mostly restating basic ideals and ideas already covered, the authors should consider highlighting important recommendations for practice and policy implications, relating them to the problem statement described in the essay.

Author Response

See attached PDF file.
